# Glycemic Index and Insulinemic Index of Foods: An Interlaboratory Study Using the ISO 2010 Method

**DOI:** 10.3390/nu11092218

**Published:** 2019-09-13

**Authors:** Thomas M.S. Wolever, Alexandra Meynier, Alexandra L. Jenkins, Jennie C. Brand-Miller, Fiona S. Atkinson, David Gendre, Sébastien Leuillet, Murielle Cazaubiel, Béatrice Housez, Sophie Vinoy

**Affiliations:** 1Inquis Clinical Research, Ltd. (formerly GI Labs), 20 Victoria St., 3rd Floor, Toronto, ON M5C 2N8, Canada; twolever@inquis.com (T.M.W.); alexandrajenkins@inquis.com (A.L.J.); 2Department of Nutrition, Mondelez France R&D SAS, 6 rue Rene Razel–Batiment K, 91400 Saclay, France; 3School of Life and Environmental Sciences and Charles Perkins Centre, University of Sydney, Sydney 2006, Australia; jennie.brandmiller@sydney.edu.au (J.C.B.-M.); fiona.atkinson@sydney.edu.au (F.S.A.); 4Biofortis Mérieux NutriSciences, 3 route de la Chatterie, 44800 Saint Herblain, France; david.gendre@mxns.com (D.G.); sebastien.leuillet@mxns.com (S.L.); murielle.cazaubiel@mxns.com (M.C.); beatrice.housez@mxns.com (B.H.)

**Keywords:** glycemic index, blood glucose, insulin, methods, methodology, available carbohydrates

## Abstract

An official method for determining food glycemic index (GI) was published by the Organization for International Standardization (ISO) in 2010, but its performance has not been assessed. Therefore, we aimed to determine the intra- and inter-laboratory variation of food GI values measured using the 2010 ISO method. Three laboratories (Australia, Canada and France) determined the GI and insulinemic-index (II) of six foods in groups of 13–15 participants using the 2010 ISO method and intra- and inter-laboratory Standard Deviations (SDs) were calculated. Overall mean food GIs varied from 47 to 86 (*p* < 0.0001) with no significant difference among labs (*p* = 0.57) and no food × laboratory interaction (*p* = 0.20). Within-laboratory SD was similar among foods (range, 17.8–22.5; *p* = 0.49) but varied among laboratories (range 17.5–23.1; *p* = 0.047). Between-laboratory SD of mean food GI values ranged from 1.6 to 6.7 (mean, 5.1). Mean glucose and insulin responses varied among foods (*p* < 0.001) with insulin (*p* = 0.0037), but not glucose (*p* = 0.054), varying significantly among labs. Mean II varied among foods (*p* < 0.001) but not among labs (*p* = 0.94). In conclusion, we found that using the 2010 ISO method, the mean between-laboratory SD of GI was 5.1. This suggests that the ISO method is sufficiently precise to distinguish a mean GI = 55 from a mean GI ≥ 70 with 97–99% probability.

## 1. Introduction

The glycemic index (GI) was developed in 1981 as a way to classify carbohydrate-rich foods according to their postprandial glycemic impact [1]. Evidence is accumulating that GI is a marker of carbohydrate quality relevant to public health [2]. In particular, a strong case can be made that high GI diets are causally related to the development of type 2 diabetes [3,4]. There are many real and perceived barriers to the practical application of GI [5]; however, one real requirement is the need for an accurate and precise method for measuring GI. In 1997, a Joint FAO/WHO consultation clarified the procedures for measuring GI [6], because it was recognized that the methodology for determining GI affects the result. In 2005, the method was reviewed and updated [7], and in 2010, an official method for determining GI was published by the Organization for International Standardization [8]. No fundamental changes from the original method were recommended in 2005 and 2010, but additional procedures were introduced to improve accuracy and precision.

The accuracy and precision of GI methodology has been assessed in two international interlaboratory studies, the first involving 7 centers [9] and the second 28 centers [10]. The procedures followed in these studies were in line with the 1997 FAO/WHO recommendations [11], and the results of both suggested that the inter-laboratory SD of GI values is approximately nine. These studies identified several issues which may affect the accuracy and precision of GI testing, and the lessons learned were incorporated into the ISO method [10]. However, the performance of the ISO method has not been tested.

Since hyperinsulinemia is associated with obesity and insulin resistance [11], it is often considered that high postprandial insulin responses are undesirable. Therefore, Health Canada and the European Food Safety Authority (EFSA) require that for a food to carry a claim related to a reduced glycemic response, it must be shown that its insulinemic response is not disproportionately increased [12,13]. This raises the question as to the accuracy and precision of the methods used to measure insulinemic responses and insulinemic index (II). Therefore, the objective of the present study was to compare the GI and II values of different food products measured in three laboratories using the current ISO method for determining the GI of foods. Since several groups have raised concerns about large inter-assay variation among different commercial insulin kits [14,15], each laboratory used the same kit for insulin measurement.

## 2. Materials and Methods

### 2.1. Participants and Procedures

Three laboratories with experience in GI determination participated in the study: Sydney University, Australia; GI Labs in Toronto, Canada and Biofortis Mérieux NutriSciences in Saint-Herblain, France. All laboratories applied the same study protocol. The study was registered at clinicaltrials.gov website with identifier NCT01870570.

At least 15 normal, healthy participants were recruited from volunteer databases. Eligible participants were men and women, 18–35 years old and non-smokers, with Body Mass Index (BMI) between 19 and 25 kg/m², normal fasting plasma glucose < 5.6 mmol/L [16] and without insulin resistance (HOMA-IR < 1.70). Participants also had to have a normal complete blood count, liver function tests < 1.5 times the upper limit of normal, fasting LDL-cholesterol < 5.00 mmol/L, HDL-cholesterol > 1.03 mmol/L (males) or >1.29 mmol/L (females), triglycerides < 1.70 mmol/L, systolic blood pressure < 130 mmHg, and diastolic blood pressure < 85 mmHg. The main exclusion criteria were inflammatory or metabolic diseases, prescribed medication at the time of inclusion that may interfere with carbohydrate metabolism, restrictive or specific diet, food allergies or hypersensitivities.

In each laboratory, participants were studied on nine separate occasions: six occasions on which they consumed different test-foods and three occasions (first, fifth and ninth sessions) were dedicated to the reference food (glucose solution containing 50 g glucose). On the morning of each test session, participants reported to the laboratory after an overnight fast (at least 10 h). Two capillary blood samples were collected by finger-pricking (hand heated in very hot water bath or with heating pads to increase blood circulation) at −10 and −5 min; the average of the glucose concentration at these two time-points was taken to be the baseline (fasting) concentration. Then participants were served a test-food together with 250 mL water. They were instructed to consume all of the food and water at a comfortable pace within 12 min. Further finger-prick blood samples were collected at 15, 30, 45, 60, 90, and 120 min after starting to eat.

The test-foods consisted of six commercial cereal products covering a wide range of expected GI: corn flakes, gingerbread, cracker, white bread, rotary-molded biscuit (RM biscuit) and sandwiched rotary-molded biscuit (SRM biscuit). For each test-food, the same batches were provided to the three laboratories. Their nutritional composition was analytically measured (fibres: AOAC 985-29; total starch: enzymatic method as described in the French standard V18-121; mono- and disaccharides were measured by High Pressure Ions Chromatography; fat and moisture: methods described in the French Decree of 08 September1977; proteins: Kjeldahl method). In-vitro starch digestibility parameters were assessed using the method developed by Englyst et al. [17]. The available carbohydrate content was calculated according to the formula provided by Brouns et al. [7]. The test-foods were served to the participants in portions containing 50 g of available carbohydrates. The macronutrient composition of the 50 g of available carbohydrate portions is shown in Table 1. The reference food was prepared by dissolving 50 g of anhydrous glucose or 55 g dextrose in 250 mL plain water. The six test-foods were provided to each participant in a random order in between the three reference food tests, according to the randomization list established prior the start of the study. The same list of randomization was used in the three labs.

Blood samples were collected in a micro tube containing heparin sodium salt (anticoagulant) for insulin quantification or sodium fluoride plus potassium oxalate (anticoagulant plus preservative) for blood glucose measurement. Blood for insulin analyses in the three laboratories and for glucose analyses in two laboratories were centrifuged and the plasma transferred into labeled, uncoated plastic micro tubes and stored at −20 °C until analyzed (<1 month from sample collection). In two laboratories, plasma glucose was measured in duplicate by enzymatic determination (glucose hexokinase/glucose-6-phosphate dehydrogenase enzymatic assay or GOD-PAP method) using a Roche Hitachi 912^®^ chemistry analyzer (Boehringer Mannheim Gmbh, Mannheim, Germany). In the other laboratory, whole blood glucose analysis was done using a YSI model 2300 STAT analyzer (YSI Inc, Yellow Springs, OH, USA). Given the lack of homogeneity reported on methodologies and assays for insulin dosage [14,15], all centers used the same ELISA kit to measure insulin concentrations (ALPCO, Salem, MA, USA) with internal standards and controls in the three laboratories. All of the blood samples collected from a single participant at one test session were analyzed within the same assay run for glucose and insulin.

The study was performed in accordance with the revised Declaration of Helsinki and Good Clinical Practice (CPMP/ICH/135/95), with European regulatory requirements (Directive 75/78/CE), and with Australian National Health and Medical Research Council (NHMRC) guidelines. The study protocol was approved by the relevant Ethics Committees for each site and in France by the French National Agency for the security of drugs and health products (Agence Nationale de Sécurité du Médicament et des produits de santé, ANSM). Written informed consent was obtained from all participants after being provided with oral and written information about the protocol study.

### 2.2. Data Analysis

The inclusion of 10 or more participants is recommended to determine the GI of food products [6,8]. To reduce the 95% margin of error of each estimate, 15 eligible healthy participants were enrolled per laboratory. Any participant who prematurely withdrew from the study was replaced in order to have at least 15 evaluable participants at the end of the trial. Therefore, some products in the ITT were evaluated on more than 15 participants per lab.

The ISO method requires, for a valid GI measurement, that the mean within-individual coefficient of variation of glycemic responses elicited by repeated tests of oral glucose (termed reference CV) is ≤30% [8]. We calculated reference CV for glucose and insulin according to the ISO method; namely the mean, SD and CV (100 × SD/mean) of the glucose and insulin iAUC values elicited by the three repeated tests of 50 g glucose were calculated for each participant and the mean of the resulting values was the reference CV. Intra-laboratory variability of the reference food was also investigated in each laboratory by partitioning the variance into among-group and within-group components and calculating the inter- and intra-individual coefficient of variation (CV).

For glucose and insulin, an incremental area under the curve (iAUC) between 0 and 120 min was calculated using the trapezoidal rule ignoring the area below the baseline. The fasting concentration used to calculate increments was defined as the average between the two fasting values at −10 and −5 min [8]. The GI and II elicited by each test-food in each participant were calculated by dividing the participant’s respective mean iAUC value for glucose or insulin after ingestion of a food portion providing 50 g of available carbohydrates by each individual’s respective mean iAUC value for glucose or insulin after ingestion of the reference food multiplied by 100. Within each laboratory, individual GI or II values greater than 2 SD above the mean were considered as outliers and were excluded from calculation of average GI and II [8,10]. The mean of the resulting values for each food was its GI or II. The primary endpoint was GI. Secondary endpoints were II, iAUC for glucose and insulin, the peak value for glucose and insulin concentration achieved (Cmax), and the peak rises of glucose and insulin (delta peak obtained by difference between fasting concentration and Cmax). All statistical analyses were performed using SAS^®^ software version 9.3 (SAS Institute Inc., Cary, NC, USA).

Assumptions of normality and homoscedasticity (homogeneity of variances) were investigated respectively by the Shapiro-Wilk and Levene tests for each study outcome. GI, II and peak rise were analyzed by analysis of variance (ANOVA) for repeated measures examining for the main effects of test-food and laboratory and the test-food × laboratory interaction. iAUCs and Cmax of glycemia and insulinemia were analyzed using an Analysis of Covariance (ANCOVA) for repeated measures to evaluate test-food and Laboratory effects, and the test-food × laboratory interaction, adjusting for Baseline after the demonstration of significant heterogeneity; individual means were compared using Tukey adjusted pairwise comparisons. Between-lab variation was assessed as the SD and CV of the three mean GI values for each food.

Statistical analyses for GI and II were performed for all participants who tested the reference food (glucose) at least two times as required by the ISO method [8]. For the other parameters, statistical analyses were performed on both intent-to-treat (ITT) and per protocol (PP) populations. If not stated differently, analyses performed on the ITT population were described in the present paper. For all statistical tests, a *p*-value < 0.05 was considered statistically significant except for normality and homoscedasticity tests for which *p*-value < 0.01 was considered as significant. Unless otherwise indicated, data are presented as means ± SD.

## 3. Results

### 3.1. Study Population

A total of 47 participants were recruited (15 in Lab 1, 16 in Lab 2, and 16 in Lab 3). The gender distribution, age, BMI, fasting plasma insulin concentrations, and HOMA-IR at baseline did not differ significantly among laboratories. Fasting glucose was significantly lower in Lab 2 than the other centers, which is likely due to the fact that Lab 2 measured whole blood glucose while the other centers measured plasma glucose (Table 2).

Five participants dropped out of the study prematurely (*n* = 2 in Lab 2 and *n* = 3 in Lab 3). The ITT population was defined as all participants enrolled in the study who were randomized and consumed at least one of the test products (*N* = 47), whereas the PP population included participants of the ITT population presenting no major protocol deviations (*N* = 42). All the data presented in this publication refer to the ITT population.

### 3.2. Glycemic Index and Glycemic Response Parameters

Mean GI values differed significantly among foods, but not among labs, and there was no significant test-food × laboratory interaction (Table 3). Gingerbread (mean ± SD GI = 86 ± 23) was classified as high-GI (>69) in all three labs while Rotary-molded biscuit (GI = 47 ± 18) was classified as low-GI in all three labs. Corn flakes (GI = 74 ± 21) and White bread (GI = 68 ± 22) were classified as having a high-GI by some labs and medium-GI (GI between 56 and 69) by others; Cracker (GI = 57 ± 18) and Sandwiched rotary-molded biscuit (GI = 50 ± 19) were classified as having a low-GI by some labs and a medium-GI by others. No food was classified as high-GI by one lab and low-GI by another. The 95% confidence interval did not include 70 for any food with a low-GI, and the 95% confidence interval did not include 55 for any food with a high-GI (Figure 1).

Intra-laboratory variation of GI (SD of GI values) did not differ significantly among foods (*p* = 0.45) but varied among laboratories (*p* = 0.056), with Lab 3 having a higher SD than Lab 2 (Table 3; Appendix A). Mean intra-laboratory GI was positively correlated with intra-laboratory SD (Figure 2A) with each 10-unit increase in GI associated with a 1.4 unit increase in SD. The mean inter-laboratory SD (SD of lab means) was 5.1 with a mean CV of 8.1% (Table 3, Appendix A). Inter-laboratory SD tended to increase as GI increased, but the correlation was not significant (Figure 2C).

Mean iAUC for glucose differed significantly among test-foods, but there was no significant laboratory effect and no test-food × laboratory interaction (Table 4). Significant differences among test-foods were as follows: glucose > all six test-foods; gingerbread > all other foods except corn flakes; corn flakes > crackers, RM and SRM biscuits and white bread > SRM and RM. There were significant differences among test-foods for glucose Cmax and peak rise but no significant effect of laboratory and no test-food × laboratory interaction (Appendix A). Glucose response curves are shown in Appendix A.

The variability of the three repeats of the reference food (refCV) was investigated within and between individuals in each laboratory on iAUC values (Table 2). The mean within-individual refCV for glycemia differed in the three laboratories, ranging from 18.0 to 29.7%. They were all below the 30% threshold stipulated in the ISO method to reduce within-individual variability of AUC and improve accuracy of GI. Similar narrow differences were observed for the between-individual variability of iAUC (0–120 min) of glycemia.

### 3.3. Insulinemic Index and Insulin Response Parameters

Mean II values differed significantly among foods (*p* < 0.001) and laboratories (*p* = 0.0037) and there was a significant test-food × laboratory interaction (Table 3, Figure 1; *p* = 0,014). This showed a higher II for corn flakes in Lab 1 compared to Lab 3 (Lab 1: 80 ± 21 vs. Lab 3: 56 ± 11). Intra-laboratory variation of II (SD of II values) did not differ significantly among foods or among laboratories (Table 3; Appendix A). There was a strong correlation (r = 0.719, *p* = 0.0008) between mean and SD of II, with every 10-unit increase in mean II being associated with an increase in SD of 4.8 (Figure 2B). The mean inter-laboratory SD (SD of lab means) was 7.8 with a mean CV of 11.6%. There was no correlation between inter-laboratory SD and mean II (Figure 2D).

Mean iAUC for insulin differed significantly among test-foods and laboratories, but there was no significant test-food × laboratory interaction (Table 4). Glucose elicited a significantly higher insulin iAUC than all the other test-foods. Gingerbread led to a greater value than SRM and white bread led to a higher iAUC value than SRM and RM. Mean insulin iAUC was significantly higher in Labs 2 and 3 than Lab 1. Mean insulin Cmax and peak rise differed significantly among test-foods and among laboratories. There was no significant test-food × laboratory interaction for either Cmax or peak rise (Appendix A). Insulin response curves are shown in Appendix A.

## 4. Discussion

The results show that there was no significant difference in the mean GI of six foods covering a wide range of GI values measured using the ISO method [8] among the three participating laboratories, and no significant food × laboratory interaction. The mean between-laboratory SD of GI values was 5.1 with a CV of 8.1%. These results suggest that the ISO 2010 method for measuring GI may result in less between-laboratory variation than earlier methods which yielded a between-laboratory SD of 9 [5,6] and is precise enough to distinguish between low-GI and high-GI foods.

The ISO method [8] incorporates six requirements, which for some of them were not found in earlier methods [6,7], intended to improve the accuracy and precision of the results of GI testing, namely, including at least *n* = 10 participants, each of whom must perform at least two tests of the reference food, the collection of two fasting blood samples, use of a glucose analytical method with CV < 3.6%, the mean within-individual variation of iAUC elicited by the reference food must be <30%, and outliers (defined as values outside the range of mean ± 2SD) should be excluded. The mean inter-laboratory SD we found here, 5.1, appears to be much less than that from previous inter-laboratory studies [9,10] of about 9. However, since the performance of different labs varies, it may only be valid to compare the SD of inter-laboratory variation from the previous study [10] for the three laboratories which participated in the present study. When this is done, the mean inter-laboratory SD here, 5.1, is somewhat less than that for the same laboratories in the previous study, 5.9 (Figure 2C). It is of interest that the intra-laboratory SDs found here for each lab were similar to those from the previous study, and the relationships between SD and mean GI were almost identical (Figure 2A).

The magnitude of inter-laboratory variation is useful for verifying the accuracy of a published or labelled GI value; a between-laboratory SD of 5.1 means that the 95% confidence interval for the difference between the two mean values will be about ±14 (assuming that the ISO method was used for both measurements and exactly the same food, prepared in the same way was tested). Alternatively, it can be used to determine the probability that the difference between two (or among >2) GI values is due to random error or to true differences in testing methodology or the food tested.

By contrast, the magnitude of intra-laboratory SDs, along with the number of participants included, is used to determine the 95% confidence limits of each mean GI. Food GI values are categorized as being Low- (≤55), Medium- (56–69) or High-GI (≥70). The mean intra-laboratory SDs for the three labs here varied from 17.5 to 23.1; over this range of SD, if the mean GI of a food measured in *n* = 10 participants was 55 (upper limit of the low-GI category), the chance that the real GI of the food was ≥70 (High-GI) would range from 1.6 to 4.4%. This suggests that the 2010 ISO method [8] is reliable for categorizing the GI values of foods. However, the reliability of the GI methodology was questioned recently by Matthan et al. who concluded that there is substantial variability in individual responses to GI value determinations, demonstrating that it is unlikely to be a good approach to guiding food choices [18]. Matthan et al. measured the GI of a commercial white bread in 63 volunteers and obtained a SD of 15.3, lower than 16 of the 18 SDs we report here (Table 3). The reason for the lower SD is likely that the GI in each participant was calculated from the means of three repetitions of glucose (reference) and three repetitions of bread, whereas we tested the reference food (glucose) three times, but each test-food only once. Matthan et al.’s conclusions do not recognize that the purpose of GI is to measure the capacity of dietary-available carbohydrates to elevate blood glucose nor the fact that glycemic responses vary greatly from day-to-day within participants; that is why it is necessary to measure the GI in at least *n* = 10 participants (i.e., the mean of at least 10 replicates) in order to obtain a reliable value. However, in showing that the GI value obtained is affected by the blood sampling schedule and method of calculating iAUC, Matthan et al. support the ISO method by confirming the need to define these procedures precisely. In addition, Matthan et al. showed that neither the mean nor the SD of the GI were affected when the number of subjects was increased from 10 up to 63. This supports the reliability of the ISO method, which specifies that a minimum of *n* = 10 subjects is required to measure GI [8] and is sufficient to yield results which are unlikely to mis-classify a High-GI food as being Low-GI, and vice-versa.

A feature of this protocol which went beyond the ISO method concerned the more restrictive inclusion/exclusion criteria for participant eligibility. It is recommended that GI testing be carried out in adults without diabetes [7,8], with no consideration of age, sex, ethnicity, or BMI in order to facilitate participant recruitment and so that the results are applicable to the general healthy population. The reason for this is that although factors such as age, sex, BMI, ethnicity, insulin resistance, and glucose tolerance status influence glycemic responses, they do not influence the GI of foods [10,19,20,21,22]. Although the GI values of foods are not affected by the presence of diabetes, people with diabetes are excluded from routine GI testing because a longer period of blood sampling is generally used for GI testing in people with diabetes [22]. Nevertheless, concerns are still raised that the variability of GI values might, at least in part, be due to differences in insulin sensitivity among participants [22]. Therefore, participants in the present study had to be lean and insulin-sensitive (evaluated as HOMA-IR < 1.7) with no features of metabolic syndrome. Unfortunately, we did not simultaneously test groups of overweight, insulin-resistant participants, so direct comparison cannot be made. Nevertheless, the fact that intra-laboratory variation was similar to the previous studies which had more inclusive inclusion criteria [10,19] suggests that restrictive inclusion/exclusion criteria may not influence the results of GI testing.

There has been considerable interest in determining the II of foods [23,24,25] and validating the II for application to mixed meals [26]. In the present study, we chose to use the Insulin ELISA kit from Alpco Diagnostics (Salem, MA, USA) because of its acceptable intra- and inter-assay variations and low interference with human pro-insulin [27]. The fact that insulin assays are less precise than glucose assays may have contributed to the higher total variation of insulin iAUC compared to glucose iAUC (Table 4, Appendix A). However, most of the variation of insulin iAUC was accounted for by between-individual variation, which was three times that for glucose iAUC. This is due to the fact that insulin sensitivity varies widely in individuals with normal blood glucose concentrations and that variations in insulin sensitivity are compensated for by variation in postprandial insulin response in order to maintain normal blood glucose [28]. Even though we only included insulin-sensitive individuals (HOMA-IR < 1.7), HOMA-IR still varied over a >three-fold range (0.49 to 1.68).

Since blood glucose concentration is the primary stimulus to insulin secretion, it is not surprising that the mean II values were significantly correlated to the mean GI values (Appendix A). However, the II values tended to be higher than the GI values with the absolute difference between II and GI being greater than the 95% margin of error (95%ME) for both II and GI for eight of the 18 pairs of values. This may be because slowly digestible starch (SDS), fibers, protein, and fat influence glucose and insulin responses [29,30]. In 190 cereal products varying from 0.1–18 g (median 10 g) SDS, 0.5–18 g (median 10 g) fat, and 0.6–30g (median 3.6 g) dietary fiber per 50 g avCHO, the effect of fat was found to interact with that of dietary fiber in products with low and medium SDS contents [30]; however, this probably does not confound the results of the present study because of the low fiber content of the test-foods (Table 1), and SDS appears as the main parameter influencing the GI of products. Adding protein or fat to glucose reduces the glycemic response in part by increasing the insulin response [29], thus, the protein and fat contents of the test-foods would tend to reduce their glycemic impact relative to oral glucose. If the GI values of the test-foods are adjusted for their contents of fat and protein as previously described [31], the difference between II and GI values (II-GI) is reduced from 5.1 ± 4.8 to −0.8 ± 3.8, and significantly fewer of the differences between II and GI are greater than the 95% margin of error (95% ME) for both II and GI (1 of 18 vs. 8 of 18, *p* = 0.008 by Fisher’s exact test, Appendix A). We found previously that II values were significantly higher in participants with type 2 diabetes than those without [22], suggesting that II may only be applicable to participants without diabetes. There was no correlation between mean II and HOMA-IR of the individual participants in the present study.

## 5. Conclusions

In conclusion, using the 2010 ISO method, we found a mean between-laboratory SD (CV) of GI values of 5.1 (8.1%). The ISO method as published is precise enough to distinguish a low-GI food (reported mean = 55) from a high-GI food (real mean ≥ 70) with 97–99% probability. Between- and within-laboratory variation for II values were higher than those for GI despite the harmonization of procedures in the three laboratories.

## Figures and Tables

**Figure 1 nutrients-11-02218-f001:**
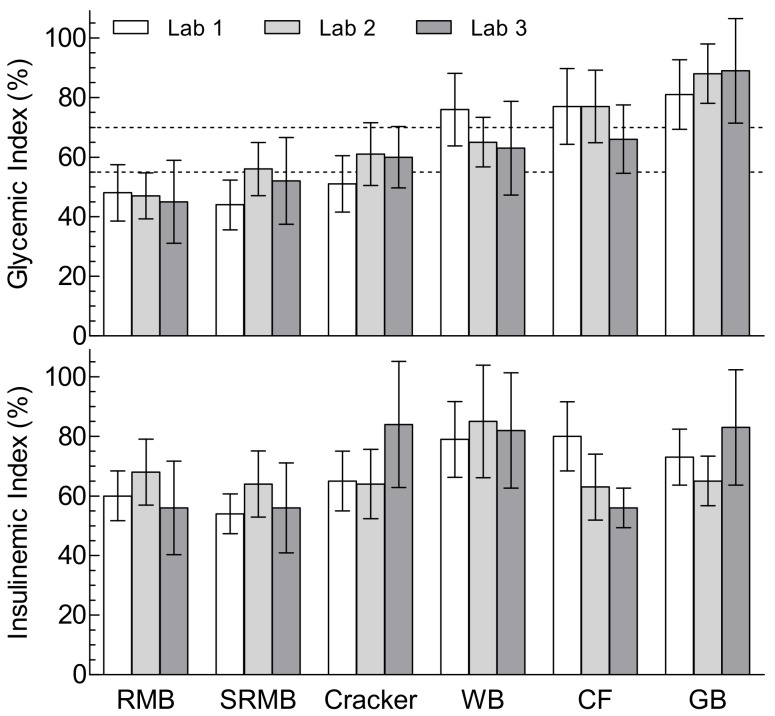
Glycemic index and insulinemic index values for the six foods in the three laboratories. Values are means ± 95% confidence intervals for *n* = 12–15 participants in each laboratory for glycemic index (top panel) and insulinemic index (bottom panel). Dashed lines on the top panel are drawn at GI = 55 (upper limit of low-GI category) and GI = 70 (lower limit of high-GI category).

**Figure 2 nutrients-11-02218-f002:**
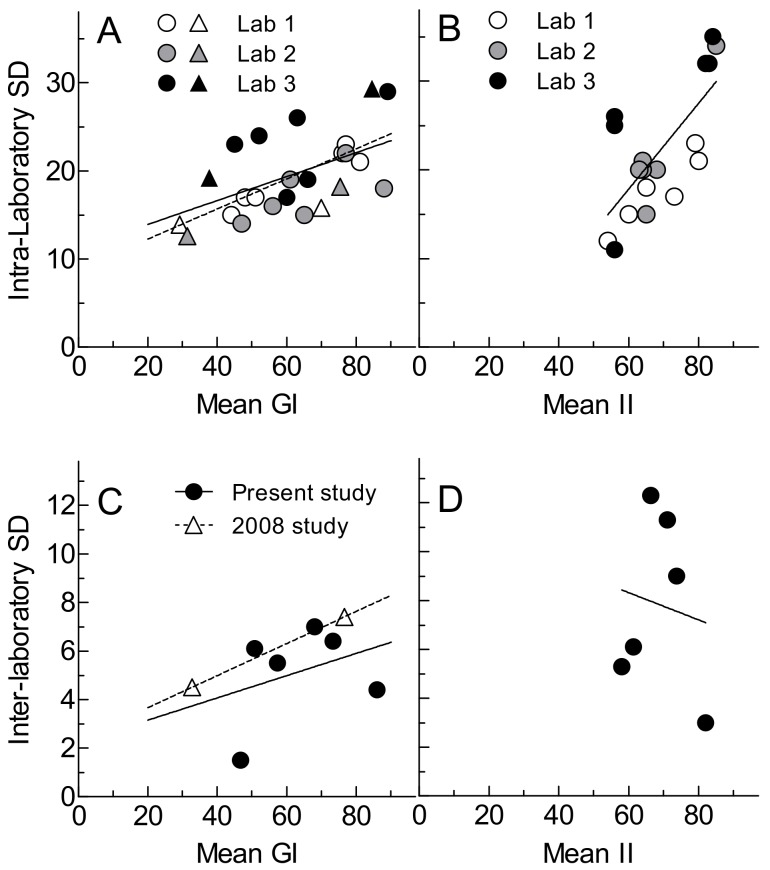
Relationships between mean and SD of GI and II within- and between-laboratories. Top panels: relationship between mean and SD of GI (**A**) and II (**B**) values of foods. Bottom panels: relationship between mean and inter-laboratory SD of GI (**C**) and II (**D**) values of foods. Circles are data from Table 3; triangles are data for the same three laboratories taken from a previous inter-laboratory study [10]. Lines are regression lines: solid lines are for data from the present study (panel A, r = 0.480, *n* = 18, *p* = 0.044; Panel B, r = 0.719, *n* = 18, *p* = 0.0008; panel C, r = 0.340, *n* = 6, *p* = 0.51; panel D, r = −0.270, *p* = 0.80). Dashed lines are for data from the previous study (panel A, r = 0.701, *n* = 6, *p* = 0.12; panel C, the line connects two points).

**Table 1 nutrients-11-02218-t001:** Composition of study products.

	Rotary-Molded Biscuit	Sandwiched Rotary-Molded Biscuit	Cracker	White Bread	Corn Flakes	Ginger-Bread
Portion weight	67.4	63.8	75.9	94.8	53.5	68.2
Moisture	1.6	1.2	1.0	29.3	1.9	11.9
Protein	4.7	4.4	5.7	9.0	4.0	2.0
Fat	11.5	11.1	17.1	4.4	0.4	1.2
Total sugars	21.2	20.5	5.8	7.0	5.0	30.2
Monosaccharides	3.0	2.5	0.7	1.9	1.6	25.4
Disaccharides	18.2	18.1	5.1	5.1	3.4	4.8
Available starch	25.4	26.0	40.0	38.9	40.8	17.7
Slowly digestible starch	11.0	8.2	4.5	0.3	1.6	0.1
Available carbohydrate ^1^	50.0	50.0	50.0	50.1	50.0	50.0
Dietary fiber	3.3	2.0	1.6	1.9	1.2	1.7

Values are grams contained in the portions fed to participants. ^1^ AvCHO calculated as follows: 1.1 × (Available starch) + 1.05 × (Disaccharides) + Monosaccharides [7].

**Table 2 nutrients-11-02218-t002:** Baseline characteristics (means ± SD) of participants at each laboratory.

	Lab 1 (*n* = 15)	Lab 2 (*n* = 16)	Lab 3 (*n* = 16)
Males/Females (n)	8/7	7/9	3/13
Age (y)	24.5 ± 4.3	24.9 ± 4.9	27.6 ± 3.9
BMI (kg/m²)	22.6 ± 1.7	23.0 ± 1.8	22.0 ± 1.9
HOMA-IR	0.90 ± 0.16	0.87 ± 0.32	0.94 ± 0.30
Fasting glucose (mmol/L)	4.57 ± 0.35 ^a^	4.08 ± 0.35 ^b^	4.67 ± 0.31 ^a^
Reference CV (glucose) ^1^	14.5 ± 6.0 ^b^	23.0 ± 11.0 ^a,b^	28.0 ± 14.2 ^a^
Intra-individual CV (glucose) ^2^	18.0	24.7	29.7
Inter-individual CV (glucose) ^2^	28.1	19.0	23.9
Fasting insulin (mIU/L)	4.40 ± 0.63	4.81 ± 1.74	4.54 ± 1.43
Reference CV (insulin) ^1^	17.1 ± 9.9	26.3 ± 14.4	24.5 ± 14.3
Intra-individual CV (insulin) ^2^	22.2	43.8	27.0
Inter-individual CV (insulin) ^2^	40.8	57.8	40.5

^1^ Within-individual variation of iAUC elicited by repeated tests of glucose calculated according to ISO 2010 method; i.e., mean of individual participants’ coefficients of variation (CV = 100 × mean/SD). ^2^ Estimate from ANOVA for repeated glucose tests. ^a,b^ Means sharing the same letter superscript do not differ significantly, *p* < 0.05.

**Table 3 nutrients-11-02218-t003:** GI and II values of the study products as determined in each laboratory.

**Glycemic Index (%)**					
**Food**	**Lab 1 (*n* = 15)**	**Lab 2 (*n* = 15)**	**Lab 3 (*n* = 13)**	**Overall (*n* = 43)**	**SD (CV) of Lab Means**
RM Biscuit	48 ± 17	47 ± 14	45 ± 23 ^1^	47 ± 18 ^1,e^	1.6 (3.3)
SRM Biscuit	44 ± 15 ^1^	56 ± 16 ^2^	52 ± 24 ^1^	50 ± 19 ^4,d,e^	5.8 (11.4)
Cracker	51 ± 17 ^1^	61 ± 19 ^1^	60 ± 17 ^1^	57 ± 18 ^3,c,d^	5.5 (9.5)
White bread	76 ± 22	65 ± 15 ^1^	63 ± 26 ^1^	68 ± 22 ^2,b,c^	6.7 (9.8)
Corn flakes	77 ± 23	77 ± 22 ^1^	66 ± 19 ^1^	74 ± 21 ^2,a,b^	6.4 (8.7)
Ginger-bread	81 ± 21	88 ± 18	89 ± 29	86 ± 23 ^a^	4.8 (5.6)
Mean of Means	63 ± 17	66 ± 15	63 ± 15	0.202 *	5.1 ± 1.9
Mean of SDs	[19 ± 3]	[17 ± 3]	[23 ± 4]		(8.1 ± 3.0)
**Insulinemic Index (%)**					
**Food**	**Lab 1 (*n* = 15)**	**Lab 2 (*n* = 15)**	**Lab 3 (*n* = 13)**	**Overall (*n* = 43)**	**SD (CV) of Lab Means**
RM Biscuit	60 ± 15	68 ± 20	56 ± 26 ^1^	62 ± 20 ^1^	5.8 (9.4)
SRM Biscuit	54 ± 12 ^1^	64 ± 20 ^1^	56 ± 25 ^1^	58 ± 19 ^3^	5.5 (9.6)
Cracker	65 ± 18 ^1^	64 ± 21 ^1^	84 ± 35 ^1^	71 ± 26 ^3^	11.0 (15.5)
White bread	79 ± 23	85 ± 34	82 ± 32	82 ± 29	3.3 (4.0)
Corn flakes	80 ± 21 ^a^	63 ± 20 ^1,a,b^	56 ± 11 ^1,b^	67 ± 21 ^2^	12.5 (18.7)
Ginger-bread	73 ± 17	65 ± 15 ^1^	83 ± 32	73 ± 23 ^1^	9.0 (12.2)
Mean of Means	69 ± 11	68 ± 8	70 ± 15	0.014 *	7.8 ± 3.5
Mean of SDs	[18 ± 4]	[22 ± 6]	[27 ± 9]		(11.6 ± 5.2)

Values are means ± SD. RM = rotary molded; SRM = sandwiched rotary molded. Values in () are coefficients of variation (CV) and values in [] are means ± SD of lab SDs. ^1,2,3,4^ The number in the superscript equals the number of outliers excluded. ^a,b,c,d,e^ Means not sharing the same letter superscript differ significantly (Tukey’s *p* < 0.05). A significant product effect is described in the overall GI values column and a significant Lab effect for the II of Corn flakes is also illustrated. * *p*-value for test-food × laboratory interaction.

**Table 4 nutrients-11-02218-t004:** Incremental areas under the curve (iAUC) for glucose and insulin.

**Glucose iAUC (mmol × min/L)**					
**Food**	**Lab 1 (*n* = 15)**	**Lab 2 (*n* = 15)**	**Lab 3 (*n* = 13)**	**Overall (*n* = 43)**	**SD (CV) of Lab Means**
RM Biscuit	102 ± 53	105 ± 44	127 ± 63	111 ± 8 ^e^	13.5 (12.1)
SRM Biscuit	103 ± 80	130 ± 37	148 ± 67	126 ± 10 ^d,e^	22.1 (17.4)
Cracker	119 ± 73	141 ± 42	170 ± 62	141 ± 10 ^d,f^	24.5 (17.1)
White bread	155 ± 63	150 ± 43	182 ± 72	159 ± 9 ^c,f^	12.2 (7.6)
Corn flakes	160 ± 63	174 ± 50	188 ± 84	172 ± 10 ^b,c^	12.9 (7.5)
Ginger-bread	164 ± 55	198 ± 59	230 ± 70	195 ± 10 ^b^	31.6 (16.1)
Glucose	207 ± 62	218 ± 59	268 ± 75	230 ± 10 ^a^	27.7 (12.0)
Mean of means	144 ± 38 ^b^	159 ± 40 ^a,b^	187 ± 48 ^a^	0.5829 *	20.6 ± 7.8
Mean of SDs	[64 ± 10 ^a^]	[48 ± 9 ^b^]	[70 ± 8 ^a^]		(12.8 ± 4.2)
**Insulin iAUC (nmol × min/L)**					
**Food**	**Lab 1 (*n* = 15)**	**Lab 2 (*n* = 15)**	**Lab 3 (*n* = 13)**	**Overall (*n* = 43)**	**SD (CV) of Lab Means**
RM Biscuit	6.04 ± 3.51	10.55 ± 4.75	9.58 ± 8.52	8.68 ± 6.03 ^b,c^	2.37 (27.2)
SRM Biscuit	5.92 ± 4.15	11.03 ± 6.49	8.74 ± 4.57	8.45 ± 5.51 ^c^	2.56 (29.9)
Cracker	6.91 ± 4.69	12.01 ± 9.75	12.98 ± 7.78	10.51 ± 8.06 ^b,c,d^	3.26 (30.7)
White bread	7.45 ± 3.38	14.18 ± 8.43	11.45 ± 3.94	10.95 ± 6.37 ^d^	3.39 (30.7)
Corn flakes	7.52 ± 2.88	11.36 ± 6.77	8.77 ± 3.18	9.18 ± 4.88 ^b,c,d^	1.96 (21.2)
Ginger-bread	6.84 ± 2.71	11.65 ± 6.11	12.34 ± 7.08	10.26 ± 6.06 ^b,d^	3.00 (29.1)
Glucose	9.64 ± 3.93	16.54 ± 9.53	14.84 ± 5.38	13.83 ± 7.51 ^a^	3.59 (26.3)
Mean of means	7.19 ± 1.25 ^b^	12.47 ± 2.14 ^a^	11.24 ± 2.32 ^a^	0.5125 *	2.93 ± 0.65
Mean of SDs	[3.61 ± 0.70 ^b^]	[7.41 ± 1.88 ^a^]	[6.00 ± 2.05 ^a^]		(28.3 ± 3.22)

Values are means ± SD. RM = rotary molded; SRM = sandwiched rotary molded. ^a,b,c,d,e,f^ Means not sharing the same letter superscript differ significantly (Tukey’s *p* < 0.05). * Significance of test-food × laboratory interaction.

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
