# Peer review of "Glycemic Index and Insulinemic Index of Foods: An Interlaboratory Study Using the ISO 2010 Method"

_nutrients, 2019, doi:10.3390/nu11092218_

Round 1

Reviewer 1 Report

The manuscript entitled "Glycemic index and insulinemic index of foods: an Interlaboratory study using the ISO 2010 method" is an exceptionally well-written manuscript examining practical issues applying and relating to the use of the ISO 2010 method. This manuscript adds great value to the existing literature and is extremely useful to researchers investigating the GI of foods as well as the to the scientific community involved in glucose metabolism. Excellent work.   

Author Response

Dear Reviewer 1,

Thank you very much for your review and for kind comments. Please find attached the latest version of the manuscript including the modifications done following the review process.

Best regards,

Alexandra

Reviewer 2 Report

Written as dictation from margin notes during review of the manuscript, so please excuse redundancy of points except for the fact that it probably means the point is considered important and surfaced more than once. 

In a general sense this study was well executed and serves the hypothesis tested that a standardized GI test can be used to compare foods with different CHO quality having GI between 45-85. A general weakness is that it does not place these results within the general scope and usefulness of the GI and GLoad that arises from the global impact of the published data. Some suggestions how that could be strengthened are offered below.

In terms of specifics, for a report focused on the ‘accuracy and precision’ of an assay like the GI, several inclusions of loose reporting or perspective bias occur.  For example, all subjects here are young [ave age of 23y], healthy, mostly white?  from urban, affluent settings on three different continents, but with presumably relatively uniform Western-type dietary habits. This point should be highlighted in both the abstract and the Discussion because the GI likely varies considerably in older folks, even if described as healthy, because energy metabolism shifts rather steadily from age 40y and beyond, especially in response to CHO type and quantity. Accordingly, it is likely that GI results would vary with subjects having different dietary habits as background, especially in different ethnic and cultural settings…so that the same diet CHO challenge [and a food’s GI] may very well differ based on undetected genetic influences linked to age, gender, and dietary histories that influence the microbiome.... Note for example even in this study that the greatest variation and coefficient of variation tended to be associated with lab #3, which had an unusual bias in gender with more subjects being female and slightly older. This supports the concept that the individual subject can have a major impact on the glycemic index. with This is in fact what Matthan et al were emphasizing in ref 13, whether they knew it or not,…because their population had an average age about 50y and intra-individual variation was huge on an interday basis, causing the authors to dismiss the GI as an important factor that can be generalized to everyday nutrition, ie. testing the same 50g CHO load as white bread in the same subject on different days revealed strikingly different GI of the bread meal for any given day. Obviously if you cannot generate the same GI day to day in the same individual , it makes the index somewhat dubious in terms of its application for an INDIVIDUAL reading a food label in the grocery store.

 But the point one wants to make clearly is that on a population basis, especially in young subjects who are still CHO-sensitive in that population, the G.I. has the potential for having a public health impact. I think that point needs to be clarified better in the present discussion, so the general reader is better informed.  It is the naïve, young metabolic system that is most influenced by the GI of a food, much less so for an older person who is less apt to be impacted by the GI on a food label.  At least that is my perspective at this time, knowing what I know.

This concept is not tested in the present experiment across 3 labs because all labs tested only 15 subjects [vs the 65+ in Matthan,  typically inner city ethnic minorities ]  and fed them 6 DIFFERENT foods on one occasion as a 50g glucose control meal [vs 3 times  with 1 food by Matthan]….with the GI ranging from 45-85 for those 6 foods to approximately the same degree in the 3 labs.. It is unclear that if the present study had tried to repeat their test diets on 3 occasions in the same subjects they would have come to the same conclusion, but the chances are vastly improved in the present expt because their test subject descriptors have the characteristics of a stable, predictable CHO metabolism as young, healthy controls with presumably comparable microbiomes.. It has been pointed out by others that the GI is not very useful when tested in diabetic subjects, so deviating from the standard Young, Healthy subject might lead [has lead]  to some confusion among researchers on the subject.

Thus, a potential shortcoming of the report is that it represents only young, healthy adults in three continents from affluent, urban locations that typically consume Western type diets. Whereas the glycemic response to food may be highly biased by the status of the entrenched microbiome. For example, It is not clear that similar findings would occur in subjects located in rural villages in India or Central Africa.  A comment along these lines in the discussion would seem appropriate. Thus, the real unspoken question here is whether the GI is a useful measure in older people with prediabetes or type II diabetes.

Another example of hasty reporting on line 48 is  â€˜a  SD of 9’, which is meaningless without ref to  the mean, which one assumes is  50-60 in this case?, tho not stated. It is reporting like this that loses the average reader. And if the SD was only 5-6 in this study , and for the same 3 study sites in the 28 centers reported in the 2008 GI comparison, but 9 for the entire 28 centers ,  it implies that the ‘other 25’ had substantially increased SD… So what caused that variation to expand in the other 25?...location, background diet history and by inference the microbiome, age, gender? 

The world literature suggests that total CHO intake [more than GI] is the key factor in determining risk to T2DM and chronic disease, in general. Perhaps GI applies most strongly to the young BEFORE any diabetes risk has evolved. This would be important to know and offer as a caveat. Right now, the paper is slanted towards the few researchers involved in establishing the public health implications of GI, but it may have certain limitations discussed above. The authors would do well to include the broader implications in their discussion and point to areas that need more understanding.

Looking at the composition of study products in table 1 raises the possibility that the glycemic index may represent more than just a function of total sugars and slowly digestible starch, but also be characteristic of foods that lack fat. Thus the potential exists that part of the glucose excursion following the test meal in fact has something to do with low fat intake and or essential fatty acid metabolism during that meal, which is simply exacerbated by sugar and/or extensive processing of the CHO prediet application during manufacture. I am surprised that the processing of CHO exemplified by various treatment of oats reported by these authors earlier this year is not part of the discussion.

Another shortcoming that is not unique to this study, but applies to the glycemic index testing procedure in general, is that the background history of subjects is not considered, as the dietary history presumably would have a major impact on the microbiome, which in turn would alter the glycemic response of any given meal. Again, this could be a major consideration of the differences described between Matthan et al and this study. And for comparison, why does the GI testing not include the response to a lentil-type food item, as a test food where the GI would be extremely low?

Placing the glycemic index in proper context, it is important to recognize that by and large epidemiological evidence supports the concept that foods with high glycemic index, and consumed in considerable quantity to generate a high glycemic load, contribute to the average hyperglycemia in that population; but that would be a chronic exposure, not necessarily measured with an acute meal. Although the classic emphasis for glycemic index has been centered around the carbohydrate content and composition and complexity of its manufacture, it should also consider other aspects of the meal including the protein and fat quantity and quality because these are known to influence the glycemic response. A major factor involved in the glycemic response however is the functionality of the microbiome, which in turn depends on fermentable carbohydrate in the form of certain fibers needed for us to have complete understanding of GI functionality.

In table 2 it would be useful to include the circulating fasting plasma triglyceride value on the 3 days for control OGTT, since that has major implications for metabolic syndrome and the glycemic response. This would give us more depth of understanding to relate the fasting glucose to the metabolism of the subjects that were being examined. This would be in addition to the fasting glucose already included, which of course is paramount to understanding the data.

Two points should be added to the discussion in order to reduce the bias of the importance of glycemic index.  The first would be to include a summary of the ARIC epi study  [Seidelman et al, Lancet Public Health 2018, 3, e419–e428] that demonstrates the importance of total carbohydrate percent energy of the diet, either too low [below 50% energy] or too high [above 65% energy] and either of which raises the risk of mortality without ref to GI per se, also with the added caveat that replacing carbohydrate calories with fat and protein is best accomplished from plant sources as opposed to animal products.  With these considerations the glycemic index loses some of its potential health implications/usefulness. The other point would be to add the recent Livesey the et al. publications on the major epidemiological association between glycemic index/ GLoad and type II diabetes just published in Nutrients in June 2019. They add balance to the epidemiology of GI and GLoad for global public health, again from the global public health point of view based on long term intake as opposed to individual application of the GI in persons of various ages, which has not been fully tested.

One suggestion for the discussion would be to broaden the understanding of glycemic index by pointing out that first of all it is well appreciated that intake of total carbohydrate is definitely a risk factor for diabetes type II and that what the glycemic index simply tries to do is rank the sources of carbohydrate in the diet to more clearly delineate highly processed carbohydrate and it’s sugar and starch content in contrast to the more natural forms of carbohydrate in natural forms as unrefined or complex carbohydrate in whole cereals, grains and vegetables.  A simple statement along those lines would loosen the tension associated with trying to push the glycemic index too high on the rank of things in the field of nutrition, where an obvious push back has occurred.  That is too bad, because it has its place. In fact, the nature of the products examined in this report are such that the glycemic index is narrowed to roughly 45 to 85, whereas the true range in glycemic index in foods and products globally ranges from 10 to 90 in a practical sense, so this report focuses on a narrow range of products related to the manufacture and processing of carbohydrate in common foods in industrialized nations and does not reflect the total range in available GI very well.

The methods do not reveal how lab number two, which collected whole blood only, was able to run insulins on plasma or serum. That procedure should be clarified.

In summary, the point is that host metabolic background is a key point for the GI outcome in a study like this, and GI, GLoad are probably only as good as the background diet history, which fares well when all the subject are young, healthy and examined in the context of the affluent, urban environment where a Western Diet is the typical background.  It is not so likely that mixed age and gender subjects from a rural village in India or Central Africa would give the same results because the GI depends in large part on the microbiome status of the host and is affected substantially by the factors listed abo

Author Response

Dear Reviewer 2,

Thank you for your review of the present paper and for your comments. Please find our answers in the document attached.

Best regards
